# Wealth Redistribution and Mutual Aid: Comparison Using Equivalent/Non-Equivalent Exchange Models of Econophysics

**DOI:** 10.3390/e25020224

**Published:** 2023-01-24

**Authors:** Takeshi Kato

**Affiliations:** Kyoto University Laboratory, Open Innovation Institute, Kyoto University, Kyoto 606-8501, Japan; kato.takeshi.3u@kyoto-u.ac.jp

**Keywords:** inequalities and wealth redistribution, taxes and redistribution, mutual aid, equivalent exchange, non-equivalent exchange, markets, economic flow, econophysics

## Abstract

Given wealth inequality worldwide, there is an urgent need to identify the mode of wealth exchange through which it arises. To address the research gap regarding models that combine equivalent exchange and redistribution, this study compares an equivalent market exchange with redistribution based on power centers and a non-equivalent exchange with mutual aid using the Polanyi, Graeber, and Karatani modes of exchange. Two new exchange models based on multi-agent interactions are reconstructed following an econophysics-based approach for evaluating the Gini index (inequality) and total exchange (economic flow). Exchange simulations indicate that the evaluation parameter of the total exchange divided by the Gini index can be expressed by the same saturated curvilinear approximate equation using the wealth transfer rate and time period of redistribution, the surplus contribution rate of the wealthy, and the saving rate. However, considering the coercion of taxes and its associated costs and independence based on the morality of mutual aid, a non-equivalent exchange without return obligation is preferred. This is oriented toward Graeber’s baseline communism and Karatani’s mode of exchange D, with implications for alternatives to the capitalist economy.

## 1. Introduction

Wealth inequality is a major social problem in various countries worldwide [1]. Survey results indicate that the global Gini index is approximately 0.7, indicating widespread inequality [2]. A Gini index of 0.4 represents a warning level for social unrest [3], and some countries far exceed this level, including South Africa, Namibia, and Suriname [4]. Higher social unrest leads to lower production and further inequality, which in turn leads to social unrest again, creating a vicious cycle [5].

The United Nations Sustainable Development Goals has prioritized Goal 10—which aims to “reduce income inequalities”, “promote universal social, economic and political inclusion”, and “adopt fiscal and social policies that promote equality”, among other targets—along with Goals 1, 2, 8, and 16 (no poverty, zero hunger, inclusive economic growth, and justice and inclusive institutions, respectively) [6]. Moreover, the United Nations University studies the impact of inequality on economic growth, human development, and governance, with inequality as a core concern [7]. Thus, there is a need to identify which types of economic relations—that is, which modes of exchange of wealth—result in inequality. For this, different definitions of the modes of exchange must be considered.

The economist K. Polanyi has identified three modes of economic exchange: reciprocity, redistribution, and market exchange [8]. Reciprocity includes the transfer of goods through gifts with the obligation to provide returns in non-hierarchical relationships; redistribution indicates the transfer of goods through the collection and refund of taxes based on the centrality of power; and market exchange represents the equivalent transfer of goods based on money prices in the market. In other words, reciprocity is a non-equivalent exchange with the obligation to return, market is an equivalent exchange, and redistribution is an equivalent exchange coordinated by a power center.

The anthropologist D. Graeber has presented baseline communism, exchange, and hierarchy as the three moral principles of economic relations [9]. Baseline communism is a mutual-aid human relationship wherein each person contributes based on their ability and is provided a return according to need; exchange is a process toward equivalence, an inhuman relationship that can be dissolved through profit and loss; and hierarchy represents a relationship bound and controlled by custom and precedent, with no tendency to operate through reciprocity. Therefore, baseline communism is a non-equivalent exchange without the obligation to return, exchange is an exactly equivalent exchange, and hierarchy is a specific form of redistribution with tribute imposed on proteges and alms imposed as the protection of a power center.

The philosopher K. Karatani has presented four modes of exchange as the various stages of the world system [10]. Modes of exchange A, B, C, and D represent reciprocity in civil society (gift and return), plunder and redistribution in the empire (domination and protection), commodity exchange in the capitalist economy (money and commodities), and restoration of the reciprocal-mutual aid relationship of A to a higher level in the world republic idealized by Kant, respectively. Mode of exchange A is thus a non-equivalent exchange with the obligation to return, B is a form of redistribution, C is an equivalent exchange, and D is a non-equivalent exchange without the obligation to return.

A comparison of the three typologies above show that the following definitions correspond with each other, as shown in Table 1: Polanyi’s reciprocity and Karatani’s mode of exchange A with non-equivalence and return; Polanyi’s redistribution, Graeber’s hierarchy, and Karatani’s mode of exchange B with centrality of power; Polanyi’s market exchange, Graeber’s exchange, and Karatani’s mode of exchange C with equivalence; and Graeber’s baseline communism and Karatani’s mode of exchange D with non-equivalence and without return. 

The contemporary capitalist economy and social security protections comprise a hybrid of equivalent market exchange (C) and redistribution by power center (B). In contrast, alternatives to the capitalist economy can be considered as a mutual-aid baseline communism and mode of exchange D, which sublimates mode of exchange A. Therefore, it is necessary to identify whether a hybrid of equivalent exchange and redistribution (B and C) or that of a mutual-aid non-equivalent exchange without obligation to return (D) would be preferable to suppress wealth inequality. This is also to clarify whether progressive capital taxation (a hybrid of B and C), as proposed by economist T. Piketty based on the famous inequality r (the rate of return on capital) >g (the growth rate of income) [11], or dynamic equality (similar to D), which forces the wealthy to have “skin in the game” themselves, as proposed by philosopher N.N. Taleb instead of a power-centered redistributive system [12], is preferable.

Econophysics uses a statistical, physics-based approach for examining wealth exchange and distribution and the mechanisms of redistribution (see, for example, the comprehensive reviews by Chakrabarti A. S. and Chakrabarti B. K., Rosser, and Ribeiro, respectively [13,14,15]). Champernowne explained Pareto’s law based on time series changes in income distribution through stochastic processes [16]. Angle, writing from a sociologist’s standpoint, showed that the gamma distribution arises through economic agents’ stochastic processes [17]. Furthermore, Dragulescu and Yakovenko illustrated that the monetary distribution follows an exponential Boltzmann–Gibbs distribution based on the analogy of energy conservation [18], and Chakraborti and Hayes demonstrated that a delta distribution arises when applying a model of random wealth transfer to a model of the poor and the wealthy, based on the analogy of kinetic energy exchange in collisions of ideal gas particles [19,20].

Chatterjee and Chakrabarti extended these models and showed that an exponential distribution can be obtained using a model in which wealth is randomly divided among agents [21]. Furthermore, Chakraborti and Chakrabarti indicated that gamma and power distributions can be obtained using a model in which agents follow a non-equivalent exchange, except for savings [22]. Kato et al. showed that a delta distribution can be obtained using a model in which wealth is exchanged equivalently according to the poor [23]. In addition, Guala used a non-equivalent exchange model combining exchange and tax redistribution for obtaining exponential and gamma distributions based on the tax rate [24], and Chakrabarti A. S. and Chakrabarti B. K. used a model combining non-equivalent exchange and redistribution by insurance to obtain insurance-rate-based exponential, gamma, and delta distributions [25].

Furthermore, Kato and Hiroi used a non-equivalent exchange model in which the wealthy contribute surplus stock to obtain delta- and gamma-like distributions based on the contribution rate; they showed that the contribution of surplus stock by the wealthy is necessary for activating economic flow and reducing inequality [26]. Kato further used an exchange model combining interest, profit and loss, and redistribution to obtain delta- and gamma-like distributions and demonstrated that the prohibition of interest, fair distribution of profit and loss, and redistribution based upon the quintile axiom in welfare economics are required for reducing inequality [27].

Elsewhere, Iglesias showed that inequality, as measured by the Gini index, is dramatically reduced by externally modeling the collection of a tax proportional to the wealth difference from local or global agents around the poorest agent and the redistribution of the tax to the poorest agent [28]. Moreover, Lima et al. showed that a combination of win/lose equivalent transactions based on the wealth of the poor, power-law taxes that are more burdensome on the wealthy, and tax exemptions for the poor can result in bimodal or flat wealth distributions, and that tax exemptions do not necessarily reduce inequality, as assessed using the Gini index [29]. These Iglesias and Lima models are effectively non-equivalent exchanges, since each exchange is taxed according to wealth.

The above-mentioned studies do not use models that combine an equivalent exchange with a redistribution separated from it by a certain time period, however. In this study, I aim to reconstruct an exchange model that represents a hybrid of equivalent exchange and redistribution (modes of exchange B and C) and a mutual-aid non-equivalent exchange without obligation to return (mode D) based on the above-mentioned exchange model of econophysics. I also compare redistribution and mutual aid in terms of wealth distribution, inequality, and economic flow to provide guidelines for alternative capitalism. This study is novel in that it compares redistribution with mutual-aid non-equivalent exchange. Furthermore, it describes new relationships between the following phenomena: economic flow and inequality; wealth transfer, time period, and redistribution; and surplus contribution of the wealthy, saving, and mutual aid. Based on the comparison, I provide new insights into alternatives to the capitalist economy.

In the present model (hybrid of equivalent exchange and redistribution), I use the equivalent exchange model of Kato et al. [23] to represent, in combination, Polanyi’s market exchange, Graeber’s exchange, and Karatani’s mode of exchange C, and Kato’s redistribution model [27] to represent, combinedly, Polanyi’s redistribution, Graeber’s hierarchy, and Karatani’s mode of exchange B. To model mutual-aid non-equivalent exchange, I adopt Kato and Hiroi’s surplus stock contribution model [26] to represent Graeber’s baseline communism and Karatani’s mode of exchange D, repositioning the surplus stock contribution of the wealthy as a mutual aid without the obligation of return. 

The remainder of this paper is organized as follows. Section 2 presents models of equivalent exchange and redistribution and mutual-aid non-equivalent exchange models, as well as the methods for calculating the Gini index and total exchange to assess wealth inequality and economic flow. Section 3 compares the simulation results of wealth distributions with the Gini index and total exchange calculations for the two models to illustrate their relationship. Section 4 examines the contemporary significance of mutual aid for redistribution considering these results and presents discussions on the nature of mutual aid for alternatives to the capitalist economy. Section 5 presents the key conclusions and future challenges.

## 2. Methods

### 2.1. Exchange Models

Figure 1 visualizes different exchange models that can be used to measure and understand inequality, to be explained in detail below.

#### 2.1.1. Basic Exchange Model

First, I present the basic wealth exchange model proposed by Chakraborti and Chakrabarti [22]. Two agents $i,\;j\;\left(=1,\;2,\;\cdots ,\;N\right)$ are selected randomly from among N economic agents. Let the wealth of agents i and j at time t be $m_{i}\left(t\right)$ and $m_{j}\left(t\right)$, respectively, with a common saving rate λ for both. Figure 1a shows that the two agents i and j save part of their wealth at time t with a savings rate λ and exchange the remaining wealth $\left(1-\lambda \right)\cdot \left(m_{i}\left(t\right)+m_{j}\left(t\right)\right)$, excluding savings, with a random division probability ε, which is a uniform random number defined in the range 0≤ε≤1. This basic model is a non-equivalent exchange model wherein the poor and the wealthy offer all their wealth (except their savings) in exchange. The wealth $m_{i}\left(t+1\right)$ and $m_{j}\left(t+1\right)$ of the two agents i and j, respectively, at time t+1 are expressed as
(1)$$m_{i}\left(t+1\right)=\lambda \cdot m_{i}\left(t\right)+\varepsilon \cdot \left(1-\lambda \right)\cdot \left(m_{i}\left(t\right)+m_{j}\left(t\right)\right);\;$$
(2)$$m_{j}\left(t+1\right)=\lambda \cdot m_{j}\left(t\right)+\left(1-\varepsilon \right)\cdot \left(1-\lambda \right)\cdot \left(m_{i}\left(t\right)+m_{j}\left(t\right)\right).$$

#### 2.1.2. Equivalent Exchange Model

The equivalent exchange model that matches the wealth of the poor (hereinafter, the EX model) proposed by Kato et al. [23] is based on the non-equivalent exchange model, as presented in Equations (1) and (2). As indicated in Figure 1b, the EX model determines the amount of exchange based on the wealth $\mathrm{Min}\left(m_{i}\left(t\right),\;\;m_{j}\left(t\right)\right)$ of the poorer of the two agents, i and j. The exchange amount presented by the wealthy and the poor is exchanged with a random division probability ε, which is a uniform random number in the range 0≤ε≤1. Wealth $m_{i}\left(t+1\right)$ and $m_{j}\left(t+1\right)$ at time t+1 are expressed as
(3)$$min=\mathrm{Min}\left(m_{i}\left(t\right),\;\;m_{j}\left(t\right)\right),$$
(4)$$m_{i}\left(t+1\right)=m_{i}\left(t\right)-\left(1-\lambda \right)\cdot min+2\cdot \varepsilon \cdot \left(1-\lambda \right)\cdot min;$$
(5)$$m_{j}\left(t+1\right)=m_{j}\left(t\right)-\left(1-\lambda \right)\cdot min+2\cdot \left(1-\varepsilon \right)\cdot \left(1-\lambda \right)\cdot min.\;$$

Repeating the exchange process in the EX model yields a delta distribution in which all wealth is concentrated in one agent’s hands, as shown in the literature [23]. Furthermore, in the EX model, redistribution is newly combined with the equivalent exchange shown in Equations (3)–(5). For the redistribution, I use the model proposed by Kato [27]. In this model, the wealth transfer rate ξ and the time period $t_{p}$ for redistribution are set, and N agents simultaneously distribute the wealth $\xi \cdot m_{i}\left(t\right)$ corresponding to the transfer rate ξ to all others equally in every period $t_{p}$ (Figure 1b). This is because establishing an average period and an average amount of redistribution when assessing the effectiveness of redistribution in reducing inequality is considered sufficient. The wealth $m_{i}\left(t+\Delta \right)$ of agent i at time t+Δ immediately after period $t_{p}$ is expressed as
(6)$$m_{i}\left(t+\Delta \right)=\left(1-\xi \right)\cdot m_{i}\left(t\right)+\xi \cdot \frac{\sum_{j\neq i}m_{j}\left(t\right)}{N-1}.\;$$

#### 2.1.3. Non-Equivalent Exchange Model

I use the model proposed by Kato and Hiroi [26] as a mutual-aid non-equivalent exchange model without obligation to return (hereafter, the NX model). The NX model is a compromise between the non-equivalent and equivalent exchange models presented in Equations (1)–(5), respectively. In the first, the wealthy contribute all surplus wealth except savings, which is not realistic in exchange, that is, economic transactions. In the second, the wealthy only contribute wealth equivalent to that of the poor; in the absence of redistribution, extreme inequality, such as a delta distribution, is likely. Thus, Kato and Hiroi set up a model in which the wealthy contribute a portion of their surplus wealth over that of the poor to control inequality to a practical extent.

As shown in Figure 1c, in the NX model, the wealth of the poor and the wealth difference between the poor and wealthy are $min=\mathrm{Min}\left(m_{i}\left(t\right),\;\;m_{j}\left(t\right)\right)$ and $\delta =\left|m_{i}\left(t\right)-m_{j}\left(t\right)\right|$, respectively; the poor take the surplus wealth $\left(1-\lambda \right)\cdot min$ as the exchange amount. The wealthy’s exchange amount is the wealth $\left(1-\lambda \right)\cdot \left(min+\gamma \cdot \delta \right)$; this is the sum of the poor’s surplus wealth $\left(1-\lambda \right)\cdot min$ and the wealth $\left(1-\lambda \right)\cdot \gamma \cdot \delta$, which amounts to the wealthy’s surplus wealth $\left(1-\lambda \right)\cdot max$ less $\left(1-\lambda \right)\cdot min$ multiplied by the surplus contribution rate γ. 

The poor and wealthy then exchange the amounts mutually proposed with a random division probability ε, which is a uniform random number defined in the range 0≤ε≤1. Graeber’s baseline communism is a mutual-aid relationship in which each person contributes based on their ability and each person is given according to their need, without obligation to return. Although the contribution of surplus wealth from the wealthy to the poor inherently varies based on need, the surplus contribution rate γ is set as a constant parameter to observe the general trends. The wealth $m_{i}\left(t+1\right)$ and $m_{j}\left(t+1\right)$ of two agents i and j, respectively, are expressed as
(7)$$min=\mathrm{Min}\left(m_{i}\left(t\right),\;\;m_{j}\left(t\right)\right),$$
(8)$$\delta =\left|m_{i}\left(t\right)-m_{j}\left(t\right)\right|,$$
(9)$$if\;\;\;m_{i}\left(t+1\right)\leq m_{j}\left(t+1\right),\;m_{i}\left(t+1\right)=m_{i}\left(t\right)-\left(1-\lambda \right)\cdot min+\varepsilon \cdot \left(1-\lambda \right)\cdot \left(2\cdot min+\gamma \cdot \delta \right);\;$$
(10)$$m_{j}\left(t+1\right)=m_{j}\left(t\right)-\left(1-\lambda \right)\cdot \left(min+\gamma \cdot \delta \right)+\left(1-\varepsilon \right)\cdot \left(1-\lambda \right)\left(2\cdot min+\gamma \cdot \delta \right).$$
(11)$$if\;\;\;m_{i}\left(t+1\right)>m_{j}\left(t+1\right),\;m_{i}\left(t+1\right)=m_{i}\left(t\right)-\left(1-\lambda \right)\cdot \left(min+\gamma \cdot \delta \right)+\varepsilon \cdot \left(1-\lambda \right)\cdot \left(2\cdot min+\gamma \cdot \delta \right);$$
(12)$$m_{j}\left(t+1\right)=m_{j}\left(t\right)-\left(1-\lambda \right)\cdot min+\left(1-\varepsilon \right)\cdot \left(1-\lambda \right)\cdot \left(2\cdot min+\gamma \cdot \delta \right).$$
The NX model equals the non-equivalent exchange model shown in Equations (1) and (2) when the surplus contribution rate γ=1 and the equivalent exchange model shown in Equations (3)–(5) when γ=0.

### 2.2. Evaluation Indices

#### 2.2.1. Gini Index

The Gini index g, used as a parameter for evaluating wealth inequality [30], is obtained by drawing the Lorenz curve and an equal distribution line [31]. Various proposed inequality indices are calculated from Lorenz curves [32], but the Gini index is used here because it is most common. Mathematically, the wealth $m_{i}\left(t\right)$ of the N agents at time t is ordered from the smallest to the largest; the k-th element in the ordered list $Sort\left(m_{i}\left(t\right)\right)$ is denoted by $r_{k}\left(t\right)$, and the Gini index g is calculated as
(13)$$r_{k}\left(t\right)\in \mathrm{Sort}\left(m_{i}\left(t\right)\right),$$
(14)$$g=\frac{2\cdot \sum_{k=1}^{N}k\cdot r_{k}\left(t\right)}{N\cdot \sum_{k=1}^{N}r_{k}\left(t\right)}-\frac{N+1}{N}.$$
When the wealth of N agents is perfectly equal (uniform distribution), the Gini index g=0; when all wealth is concentrated in a single agent’s hands (delta distribution), g=1. In other words, g ranges from 0 to 1. The greater the inequality, the larger the value of g.

#### 2.2.2. Total Exchange

The total exchange amount f is used to evaluate economic flow [26]. The total exchange f is the sum of the exchanges of the wealthy and poor $\left(1-\lambda \right)\cdot \left(2\cdot min\left(t\right)+\gamma \cdot \delta \left(t\right)\right)$ at time t from time t=1 to $t=t_{max}$.
(15)$$f=\frac{\sum_{t=1}^{t_{max}}\left(1-\lambda \right)\cdot \left(2\cdot min\left(t\right)+\gamma \cdot \delta \left(t\right)\right)}{2\cdot t_{max}}.$$

Furthermore, Equation (15) applies to Equations (3)–(5) if γ=1. The denominator in Equation (15), intended for normalization, is the total amount exchanged between the two agents from time t=1 to $t=t_{max}$, when the two agents exchange one amount each. The larger the total exchange f, the more active the exchange of wealth, that is, the economic flows are large, and the market is active.

## 3. Results

I first examine wealth distributions for the EX model of equivalent exchange and redistribution represented by Equations (3)–(6) and the NX model of non-equivalent exchange represented by Equations (7)–(12). Figure 2 shows a representative example of the simulated wealth distribution results. I set a savings rate of λ=0.25 because the average global savings rate relative to the gross domestic product (GDP) is approximately 0.25 [33], and a transfer rate of ξ=0.5 in the EX model because the highest inheritance tax rate in the Organisation for Economic Co-operation and Development (OECD) countries is approximately 0.5 [34,35]. 

A consideration of Figure 2(a1,a2) reveals that as the wealth distribution in the EX model approaches a power distribution, a delta distribution with an increase in the redistribution period $t_{p}=10^{4}$ to $10^{5}$ occurs—that is, inequality increases. This implies that some form of redistribution must be conducted because only equivalent exchange leads to extreme inequality, as suggested by the literature [23] with respect to regional inequality. A consideration of Figure 2(b1,b2) shows that the wealth distribution approaches a gamma-like distribution from an exponential distribution in the NX model when the wealthy’s surplus contribution rate increases from γ=0.1 to 0.5, that is, the inequality narrows. This suggests that inequality can be controlled if considerable mutual aid is provided in a non-equivalent exchange.

Next, I examine the change in the Gini index (inequality) g over time (number of exchanges) t for the EX and NX models by Equations (13) and (14). Figure 3 shows the results of these simulations. 

In Figure 3, cases ξ=0 (i.e., no redistribution in the EX model) and γ=0 (i.e., no mutual aid in the NX model) are identical; as time t passes, the Gini index approaches g=1, and all wealth is concentrated in one agent’s hands. In other words, in an equivalent market exchange, inequality can only be maximized. In the EX model with ξ=0.5, the redistribution period $t_{p}=10^{5}$ to $10^{3}$ is shortened. In the NX model, the Gini index g decreases and inequality is suppressed when the rate of surplus contribution from the rich to the poor increases from γ=0 to γ=0.5; however, γ=0.5 and γ=1 show little difference. The reason the Gini index saturates with respect to γ is presumably because the shape of the Lorenz curve itself, which calculates the Gini index g, does not change, although the wealthy and poor switch as γ increases, as discussed in the literature [26] regarding the rank correlation coefficient.

In Figure 2 and Figure 3, the savings rate λ=0.25 is held constant. Subsequently, I examine the Gini index (inequality) g by Equations (13) and (14) and total exchange (economic flow) f by Equation (15) for the savings rate λ and the redistribution parameter $\xi /t_{p}\times 10^{-3}$ of the EX model, and for the savings rate λ and the surplus contribution rate (mutual aid) γ of the NX model. The redistribution parameter $\xi /t_{p}\times 10^{-3}$ is introduced because the same inequality suppression effect is expected for an increase in the transfer rate ξ and a decrease in the period $t_{p}$; the $\times 10^{-3}$ is used for adjusting the computational orders of magnitude. Figure 4 shows the results of these simulations. The time (number of exchanges) t is set to $10^{6}$, at which the Gini index g is almost stable, as shown in Figure 3.

The consideration of Figure 4a,b reveals the same trend for both EX and NX models. In the EX model, the larger the savings rate λ, the smaller is the Gini index g (inequality is suppressed) and the smaller is the total exchange f (economic flow is reduced). Furthermore, the larger the redistribution parameter $\xi /t_{p}\times 10^{-3}$, the smaller is the g (inequality is suppressed) but the larger is the total exchange f (economic flow is activated). Figure 4b shows that in the NX model, the larger the savings rate λ, the smaller are the g (inequality is suppressed) and f (economic flow becomes stagnant). Moreover, the larger the mutual aid γ, the smaller is the g (inequality is suppressed) and the larger is the f (economic flow is activated). In other words, inequality g and economic flows f are inversely related with respect to the redistribution parameter $\xi /t_{p}\times 10^{-3}$ in the EX model and mutual aid γ in the NX model. As specific values are difficult to read in Figure 4, I examine the Gini index g for the redistribution parameter $\xi /t_{p}\times 10^{-3}$ of the EX model and the surplus contribution rate (mutual aid) γ of the NX model. Figure 5 shows the results of these simulations based on Figure 2, Figure 3 and Figure 4. 

Figure 5a,b shows that, as in Figure 4, the larger the redistribution parameter $\xi /t_{p}\times 10^{-3}$ in the EX model and the mutual aid γ in the NX model, the smaller is the Gini index g (inequality is reduced). In addition, both plots are accurately approximated by the saturation curve (dotted line in the figure) because the coefficient of determination $R^{2}$ is sufficiently large. At a global average savings rate λ=0.25 [33], the redistribution parameter and the mutual aid must be as follows: $\xi /t_{p}\times 10^{-3}\geq 0.2$ in the EX model and γ≥0.2 in the NX model, respectively, to avoid exceeding the warning level g=0.4 [3]. In other words, Figure 5 suggests that without a certain degree of redistribution or mutual aid, social unrest and disturbance will be triggered and Goal 10 of the Sustainable Development Goals to reduce wealth inequality [6] will not be achieved.

Finally, based on the inversely proportional relationship between the Gini index g and the total exchange f in Figure 4, I introduce the parameter f/g. Then, I examine the relationship of f/g to the redistribution parameter $\xi /t_{p}\times 10^{-3}$ in the EX model and to the parameter $\left(1-\lambda \right)\cdot \gamma$, comprising the savings rate λ and the surplus contribution rate (mutual aid) γ, in the NX model. I introduce the parameter $\left(1-\lambda \right)\cdot \gamma$ in the NX model because reducing the savings rate λ and increasing the surplus contribution rate γ are believed to increase the unitary exchange and mutual aid per exchange. In contrast, in the EX model, the transfer rate ξ is multiplied by the entire wealth, including savings, in every period $t_{p}$; thus, the effect of redistribution is considered independent of the savings rate λ. Figure 6 shows these simulation results. 

Figure 6a,b show that the parameter f/g increases as $\xi /t_{p}\times 10^{-3}$ and $\left(1-\lambda \right)\cdot \gamma$ are increased for the EX and NX models, respectively. Furthermore, both plots are accurately approximated by the saturation curves (dotted lines in the figure) because the coefficients of determination $R^{2}$ are larger than 0.9. The EX and NX models yield $f/g\sim 0.241\ln\xi /t_{p}\times 10^{-3}+1.48$ ($R^{2}=0.779$) and $f/g\sim 0.403\ln\left(1-\lambda \right)\cdot \gamma +1.92$ ($R^{2}=0.937$), respectively, when approximated by logarithmic curves. The NX model results indicate that the logarithmic curves can be approximated with adequate accuracy, which is consistent with the view in the literature [26]. In Figure 6, I compare the EX and NX models using saturation curves that can be accurately approximated because both have sufficiently large $R^{2}$. It is safe to say that both approximations are isomorphic and that holds.
(16)$$\frac{f}{g}\sim 2\left(1-e^{-5x}\right),$$
(17)$$x\sim \frac{\xi }{t_{p}\times 10^{-3}}\sim \left(1-\lambda \right)\cdot \gamma .$$

Therefore, the redistribution parameter $\xi /t_{p}\times 10^{-3}$ in an equivalent exchange and the mutual aid $\left(1-\lambda \right)\cdot \gamma$ that considers savings in a non-equivalent exchange yield roughly the same result with respect to the parameter f/g. The approximate equations shown in Figure 6 and Equations (16) and (17) imply that if the right side has a constant value, the Gini index (inequality) g and the total exchange (economic flow) f on the left side are inversely proportional, that is, activating economic flow will increase inequality. Additionally, it is necessary to increase $\xi /t_{p}\times 10^{-3}$ and $\left(1-\lambda \right)\cdot \gamma$ on the right side for the EX and NX models, respectively, to increase f/g on the left side (i.e., to increase the total exchange f while decreasing the Gini index g). Moreover, redistribution must either occur with a high transfer rate ξ and a short period $t_{p}$ or with a low saving rate λ and considerable mutual aid γ to simultaneously reduce inequality and stimulate economic flow.

The numerical values presented in Figure 6a indicate that the redistribution parameters $\xi /t_{p}\times 10^{-3}\sim 1$ and f/g~2 are at the saturation point of the EX model. At this point, the periods $t_{p}\sim 1000$, $t_{p}\sim 800$, and $t_{p}\sim 500$ should be set for transfer rates ξ~1, ξ~0.8, and ξ~0.5, respectively. Given the results in Figure 3, this is tantamount to redistributing wealth before wealth distribution occurs, which is not realistic. If the target is $\xi /t_{p}\times 10^{-3}\sim 0.2$, where f/g does not drop considerably on the saturation curve, $t_{p}\sim 5000$ for ξ~1, $t_{p}\sim 3000$ for ξ~0.6, and $t_{p}\sim 2000$ for ξ~0.4; this seems feasible within the range of the latter two, that is, ξ~0.5 and $t_{p}\sim 2500$.

Based on the numerical values presented in Figure 6b, the saturation point of the NX model is $\left(1-\lambda \right)\cdot \gamma =1$ and f/g~2. The savings rate λ=0 and the surplus contribution rate γ=1 should be set at this point; however, it is unrealistic for the wealthy to always contribute the entirety of their surplus wealth, and for the poor and the wealthy to always save no wealth, respectively. The latter is because they must save to maintain long-term future reserves and meet contingent expenditures attributable to disasters. If the target is $\left(1-\lambda \right)\cdot \gamma \sim 0.2$, where f/g does not drop considerably on the saturation curve, γ~1 for λ~0.8, γ~0.33 for λ~0.4, and γ~0.25 for λ~0.2; it would be feasible to achieve λ~0.3 and γ~0.28 within the range of the last two considering the global average savings rate of 0.25 [33].

Figure 7 shows the relationship between redistribution and mutual aid based on Equations (16) and (17). The circle represents the tentative target. Lengthening the period of redistribution from $t_{p}=2500$ to 5000 results in a transfer ratio ξ~1, that is, transferring all assets and further lengthening the period would no longer maintain the same f/g as the mutual aid, and this would lead to economic stagnation or widening inequality. Conversely, if the redistribution period is shortened from $t_{p}=2500$ to 1,250, 625, the transfer rate decreases to ξ~0.25, 0.125, which necessitates frequent redistributions.

## 4. Discussion

This study compared a model combining equivalent exchange and redistribution (Polanyi’s market exchange, Graeber’s exchange, and Karatani’s mode of exchange C combined with Polanyi’s redistribution, Graeber’s hierarchy, and Karatani’s mode of exchange B) and a mutual-aid non-equivalent exchange model (Graeber’s baseline communism and Karatani’s mode of exchange D). This comparison reveals that both produce the same computational interpretation of the results for wealth inequality and economic flow. Reducing inequality and stimulating economic flow requires either power-centered collection and redistribution at a high tax rate and frequency in an equivalent market exchange or a mutual-aid non-equivalent exchange without obligation of return, in which savings are kept low and the wealthy’s rate of surplus wealth contribution is high.

What does the computational similarity of authoritative redistribution and non-authoritative mutual aid imply? With respect to time t in these exchange models, a human lifetime would be considered equivalent to approximately $10^{4}$ order of magnitude (~365 days × 100 years). Therefore, a redistribution target of $\xi /t_{p}\times 10^{-3}\sim 0.2$ would mean that a tax of ~50% is levied once every few decades on all assets and not the income. The maximum inheritance tax rate in OECD countries (i.e., once in a lifetime) is 50% [34,35], which means that collection and redistribution should be conducted more frequently. Expenses to the government are ~30% of GDP [36], and the collection and redistribution of taxes by the power center is extra costly; furthermore, the institutional design creates redistribution bias, that is, inequality. 

In contrast, a mutual aid target of $\left(1-\lambda \right)\cdot \gamma \sim 0.2$ implies that the wealthy voluntarily give ~30% of their surplus stock to the poor in a single exchange, without obligation to return, assuming an average saving rate λ=0.25 [33]. Although a prescribed surplus contribution rate γ is specified when modeling a mutual-aid non-equivalent exchange, the original baseline communism or mode of exchange D only requires that mutual aid be provided as required. In addition, even if redistribution and mutual aid are “computationally” similar, they are “qualitatively” different in that redistribution is coercion-based and driven by the centrality of power, whereas mutual aid is a voluntary choice based on non-centrism and morality. Extra-personal altruism and compassion, as opposed to coercion, are believed to result in wellbeing [37]. Therefore, it is evident that a mutual-aid non-equivalent exchange without obligation to return (the alternative human economy) is preferable to redistribution by power centers in an equivalent market exchange (capitalist economy and social security).

Here, examining the mechanism of the Islamic economy is instructive. As a legal system, the Islamic economy encompasses politics, economics, and society and prohibits interest (*riba*) and speculation (*gharar*), which lead to inequality. Furthermore, it also successfully balances selfishness as the pursuit of self-interest through joint ventures (*mudaraba*), consensual contracts (*murabaha*), and futures trading (*salam*) and altruism as mutual aid through donation (*waqf*), alms (*sadaqah*), and charity (*zakat*) in an equal and non-centered community (*ummah*) under God [38,39,40]. Redistribution through various institutions according to the Islamic legal system, rather than coercion by power centers, is more like a non-equivalent exchange of mutual aid.

According to Graeber, history over the past five millennia has alternated between cycles of bullion-based monetary economies and virtual money-based credit economies [9]. The monetary economic period is generally characterized by interest-bearing debt, war, and slavery, whereas the credit economic period has witnessed a morally peaceful society. In the Middle Ages, a credit economy era that predated the modern era, moral and financial innovations emerged from the Islamic world. As the modern era transitions from a monetary economy to a credit economy, the Islamic economy could, once again, provide an alternative to the capitalist economy [41,42,43].

Kato compares the Islamic and capitalist economies from the econophysics perspective; he proposes a return to a “real transaction-based economy” rooted in nature and local communities, the promotion of a “face-to-face association economy”, and the revival of an “economy embedded in the morality of mutual aid” as guidelines for a credit economy as an alternative to capitalism [27]. He then states that the challenge in the non-Islamic world lies not in redistribution through taxes collected under centralized power but in mutual aid through one’s free choice under the community’s non-centrality and in the rebuilding of the morality of mutual aid, that is, without a specific religion.

These guidelines can be considered to be oriented toward anarchism. Anarchism is an ideology wherein individual freedom and communal solidarity are not contradictory. It seeks to build a free and equal society through mutual agreement. Graeber and Grubacic define anarchism in terms of four qualities: non-centrality, voluntary association, mutual aid, and the network model [44]. Graeber’s baseline communism and Karatani’s mode of exchange D, which are represented in this non-equivalent exchange model, are oriented toward anarchism as they both aim for a human economy in which free exchange occurs while incorporating the morality of mutual aid [45].

Deguchi, writing from the perspective of a philosopher, describes the East Asian view of the self, “Self-as-We”, which is connected to the lineage of Laozhuang and Zen thought, as opposed to the Western view of self, “Self-as-I” [46,47,48]. According to Deguchi, human beings have a “fundamental incapability” to live alone, and “Self-as-We” is a network of multi-agents—including “I”—who entrust themselves to each other. The “mixed-life society” in which “we” live is one in which different self-nomadic people interact, mingle, and remain in contact, recognizing each other’s “fundamental incapability” and sublimating it into solidarity. Deguchi’s ideology also underlies Graeber’s baseline communism, Karatani’s mode D of exchange, and the face-to-face association economy based on real transactions in the morality of mutual aid.

Another perspective is the triangle “state (public agencies)–community–market (private firms)” presented by Pestoff, writing as a political scientist [49]; the “public (state)–common (community)–private (market)” framework presented by policy scholar Y. Hiroi [50,51]; and the three pillars “state, community, market” presented by economist R. Rajan [52]. The state corresponds to Polanyi’s redistribution, Graeber’s hierarchy, and Karatani’s mode of exchange B; the community corresponds to Polanyi’s reciprocity and Karatani’s mode of exchange A; and the market corresponds to Polanyi’s market exchange, Graeber’s exchange, and Karatani’s mode of exchange C. Thus, Pestoff’s association at the center of the triangle, Hiroi’s synthesis of “public–community–private” and the departure from the local level, and the balance between Rajan’s three pillars is oriented toward Graeber’s baseline communism and Karatani’s mode of exchange D.

In recent work, Karatani notes that the mode of exchange D has emerged repeatedly through the return of mode of exchange A (reciprocity and return) at a higher level, not as a world religion such as a monotheistic religion supporting the empire but as a universal religion emerging on the periphery in defiance of the empire. He also states that because of the crises of war and depression induced by modes of exchange B (imperial plunder and redistribution) and C (money and commodity exchange), mode of exchange D will arrive “from beyond” human will and planning [53].

Historian W. Sheidel states that human history has witnessed wars, revolutions, collapse of states, and epidemics, decreasing economic inequality [54]. Currently, the world is suffering from the COVID-19 pandemic, war in Ukraine, and natural disasters and conflicts caused by the effects of global warming. Although these crises are unfortunate, they may hasten the arrival of mode of exchange D and facilitate the transition from a capitalist economy to the alternatives suggested by Graeber and Karatani.

As an aside, economist T. Piketty proposes progressive capital taxation based on the famous inequality r>g (the rate of return on capital > the growth rate of income) [11]. In terms of this study, this would correspond to imposing a progressive transfer rate on the wealthy in a combination of modes of exchange B and C. Furthermore, philosopher N.N. Taleb states that decentralization towards those who have “skin in the game” rather than a redistributive institution constituted by power centers will mitigate the inequality-generating asymmetry. He also formulates the idea that, to make society more equal, the wealthy should have “skin in the game” and risk falling out of the wealthy class (i.e., dynamic ergodicity should be restored) [12]. In this study’s context, this would encourage the wealthy’s surplus stock contribution, which may be said to be oriented toward the baseline communism and mode of exchange D.

This study is limited in that it compares general trends in redistribution and mutual aid, that the same transfer rate ξ and period $t_{p}$ is set for all agents in the EX model, and that the same surplus contribution rate γ is set for all agents in the NX model. Future analytical studies should be conducted in more detail, for example, by setting the transfer rate ξ and period $t_{p}$ in the EX model based on various social security programs and by choosing the surplus contribution rate γ in the NX model according to the ability of the wealthy and the needs of the poor. Moreover, empirical studies are needed that use real-world evidence to examine the relationship between economic flow and Gini index with respect to tax rate and frequency for the EX model, and with respect to stock and surplus contribution of the wealthy for the NX model.

In addition, this study uses a conservative model for aggregate wealth that deals only with exchange. Therefore, it does not deal with production and consumption, or interest and profit/loss in the real-world economy [15]. With respect to interest and profit/loss, there is a non-conservative model introduced by Kato in comparison of Islamic and capitalist economies [27]. In a future study, the redistribution or mutual aid of interest and profit/loss for the wealthy and the poor can be considered in such a non-conservative model.

It should be added that, although the present study used a model based on the kinetic energy exchange analogy, there is another model that uses potential function to compute probability distributions for income and expenditure [55], and a model that uses population dynamics to compute time developments for growth and inequality [56]. Future research could thus include such models that take into account the finiteness of earth’s resources and the sustainability of economy. Such non-conservative models are subject to the constraint of resource limits, however, and eventually researchers may wish to revert to a conservative model that is primarily based on exchange.

## 5. Conclusions

In this study, I develop econophysics-based exchange models for a hybrid of a market-based equivalent exchange (EX) and power-centered redistribution and a mutual-aid non-equivalent exchange (NX). I also compare redistribution and mutual aid in terms of wealth inequality and economic flow. 

Simulations conducted using these exchange models to evaluate the Gini index (inequality) g and total exchange (economic flow) f show that in both the EX and NX models, the larger the savings rate λ, the more the inequality is suppressed and economic flows stagnate. Furthermore, the larger the synthetic parameters $\xi /t_{p}\times 10^{-3}$ and $\left(1-\lambda \right)\cdot \gamma$ in the EX and NX models, respectively, the more the inequality is suppressed and economic flows are activated. I show that the EX and NX models have the same saturated curvilinear approximation equations $f/g\sim 2\cdot \left(1-e^{-5x}\right),\;x\sim \xi /t_{p}\times 10^{-3}\sim \left(1-\lambda \right)\cdot \gamma$ for these relationships. This approximate expression indicates that inequality and economic flows are inversely proportional and that the parameter x must be large to achieve both.

Although the EX and NX models are “computationally” isomorphic approximations, the NX model of mutual-aid non-equivalent exchange, is “qualitatively” preferable to the EX model, a hybrid of market equivalence exchange and power redistribution. This is indicative of Graeber’s baseline communism, Karatani’s mode D of exchange, a face-to-face association economy based on real transactions as learned from the Islamic economy, and the ideals of anarchism.

Notwithstanding the fact that mutual aid is “qualitatively” preferable to redistribution, there remain issues that are beyond the scope of this study’s econophysics-based approach: the reconstruction of a moral system in the non-Islamic world that is not based on any particular religion; the realization of a “mixed-life society” of “We” with “fundamental incapability;” and the incorporation of Graeber’s stated capitalist economic alternative and Karatani’s mode of exchange D. Future social practice activities based on philosophy, economics, and sociology should focus on addressing these issues. 

Specifically, in order to shift steadily from redistribution toward mutual aid—that is, toward Pestoff’s association and Hiroi’s synthesis of “public-community-private” described in the Discussion section—mutual-aid communities could be built through cooperatives [57] and social enterprises [58,59] using environmental, social, and governance investing [60] as well as social impact bonds [61]. Such cooperatives and social enterprises will require governmental policies that provide them with preferential taxation and financial resources. They will also need to be administrated in a way that allows for the delegation of authority and lateral support. Still, though the progress toward social innovation will always be confronted by various social challenges [62], we must nevertheless reduce inequalities. This may be achieved in the future through the fusion of human society and information systems, such as in platform democracy [63], platform cooperatives [64], and cyber-human social cooperating systems [65].

## Figures and Tables

**Figure 1 entropy-25-00224-f001:**
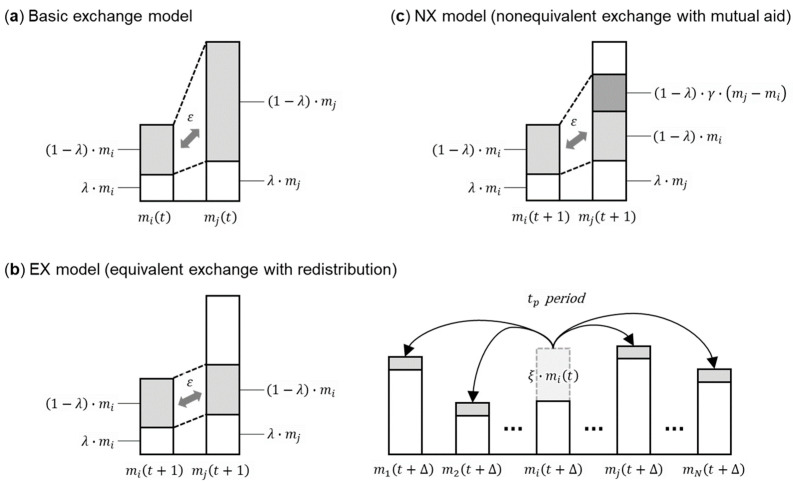
Exchange models: (**a**) basic exchange model; (**b**) equivalent exchange model (EX) with redistribution rate ξ and time period $t_{p}$; and (**c**) non-equivalent exchange model (NX) with surplus contribution rate γ. $m_{i}$ and $m_{j}$ represent the wealth of agents i and j, respectively, at times t, t+1 and t+∆. λ represents the common savings rate, and ε represents the random division probability.

**Figure 2 entropy-25-00224-f002:**
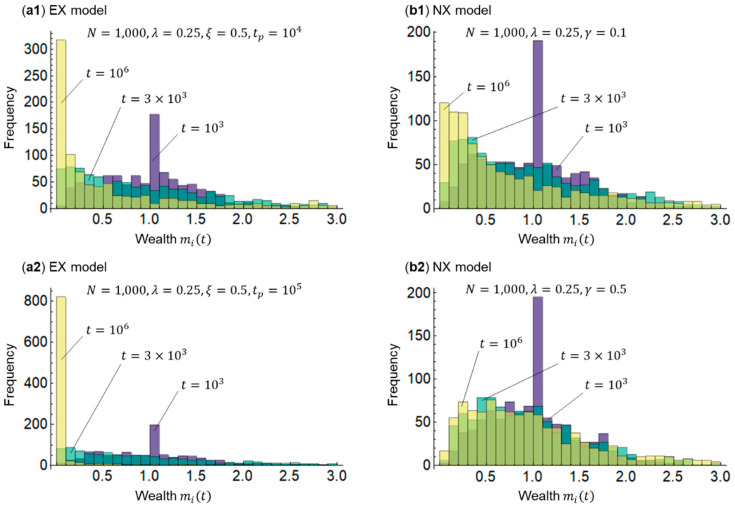
Wealth distribution. (**a1**) and (**a2**) represent EX models, and (**b1**) and (**b2**) represent NX models. In all models, the number of agents is N=1000, the initial values of wealth at time t=0 are $m_{i}\left(0\right)=1\;\left(i=1,\;2,\cdots ,\;N\right)$, and the savings rate is λ=0.25. In the EX model, the transfer rate is ξ=0.5, and the time period is $t_{p}=10^{4},\;10^{5}$. In the NX model, the surplus contribution rate is γ=0.1, 0.5. To determine the changes in wealth distribution, the time (number of exchange repetitions) is $t=10^{3},\;3\times 10^{3},\;10^{6}$.

**Figure 3 entropy-25-00224-f003:**
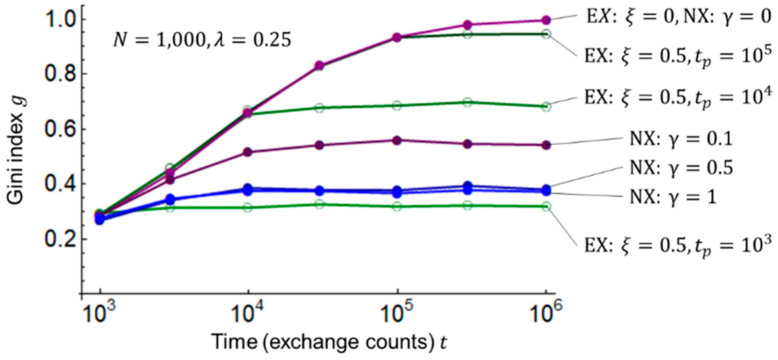
Gini index on time passage. The number of agents is N=1,000, initial values of wealth at time t=0 are $m_{i}\left(0\right)=1\;\left(i=1,\;2,\cdots ,\;N\right)$, and the savings rate is λ=0.25. In the EX model, the transfer rate is ξ=0 and 0.5, and the time period is $t_{p}=10^{3},\;10^{4},\;10^{5}$. In the NX model, the surplus contribution rate is γ=0, 0.1, 0.5, 1.

**Figure 4 entropy-25-00224-f004:**
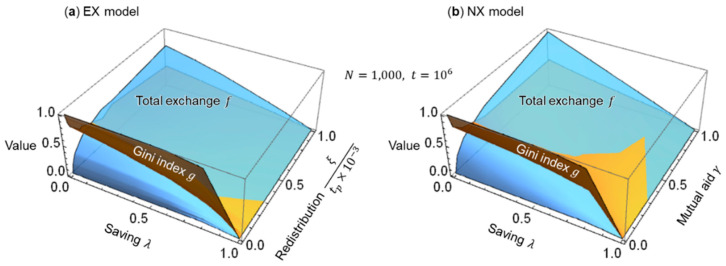
Three-dimensional graphs of Gini index g and total exchange f for saving rate λ and redistribution parameter $\xi /t_{p}\times 10^{-3}$ or mutual aid γ: (**a**) EX model and (**b**) NX model.

**Figure 5 entropy-25-00224-f005:**
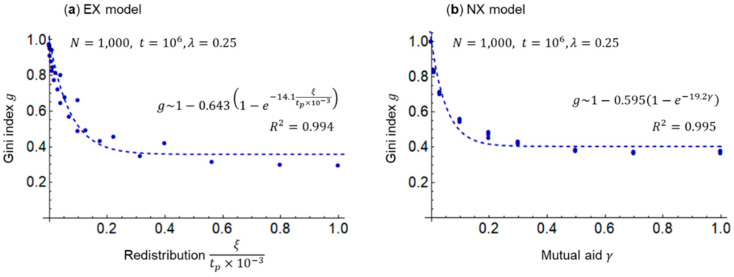
Relationship of Gini index g for the redistribution parameter $\xi /t_{p}\times 10^{-3}$ or mutual aid γ: (**a**) EX model and (**b**) NX model. In both models, dotted lines represent approximate curves.

**Figure 6 entropy-25-00224-f006:**
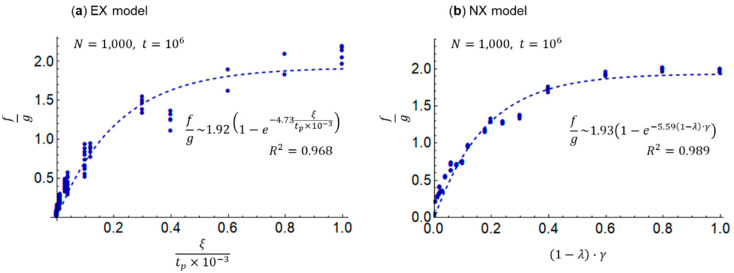
Relationship of the f/g parameter with the redistribution parameter $\xi /t_{p}\times 10^{-3}$, or mutual aid (1−λ)·γ. (**a**) EX model and (**b**) NX model. In both models, dotted lines represent approximate curves.

**Figure 7 entropy-25-00224-f007:**
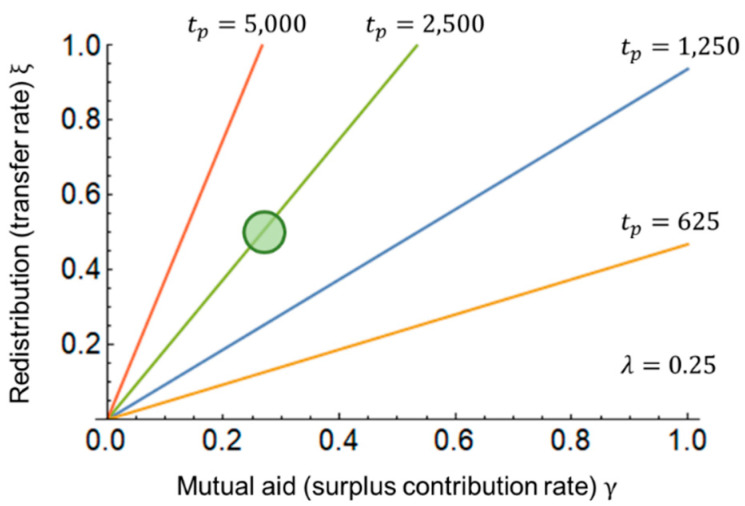
Relationship between redistribution parameter ξ and mutual aid γ. The savings rate is λ=0.25, and the time period of redistribution is $t_{p}=625,\;1250,\;2500,\;5000$.

**Table 1 entropy-25-00224-t001:** Comparison of economic typologies by Polanyi, Graeber, and Karatani.

Typology	Polanyi	Graeber	Karatani
Non-equivalent exchange with obligation to return	Reciprocity	—	Mode of exchange A
Redistribution by power center	Redistribution	Hierarchy	B
Equivalent exchange in the market	Market exchange	Exchange	C
Non-equivalent exchange without obligation to return	—	Baseline communism	D

## Data Availability

Not applicable.

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
