# Peer review of "Wealth Redistribution and Mutual Aid: Comparison Using Equivalent/Non-Equivalent Exchange Models of Econophysics"

_entropy, 2023, doi:10.3390/e25020224_

Round 1

Reviewer 1 Report

The topic is delicate but significant and the modelling extremely difficult, and the approach in the paper must recognize its limitations either from the beginning or at the end of the article.

I think some aspects must be recalled both in the introduction or literature review and in the discussions. Here I believe, obviously in my opinion, that there are certain aspects that cannot be omitted in the literature review or in discussions or conclusions. First, both Nassim Nicholas Taleb's dynamic model of real capitalist inequality and institutional Thomas Piketty's static model must be really remembered and developed as impact. The thinking that created the fragile Talebian model reaffirms truths according to which events with low probability and significant impact are not calculable, but also the static approach (does not reflect what will happen during the entire life of the individual) compared to the dynamic type (even ergodic) which takes into account the past and the future in the same measure and that brings them together as a whole. In the analysis of inequality, the thinking of Nassim Nicholas Taleb, it is found, for example, that in the USA, 39% of American individuals will be part of the first 5% in the ranking of the income distribution, 56% will be among the first 10% and 73% will be among the first 20% - Nassim Nicholas Taleb, in 2018, in Skin in the game: Hidden asymmetries in daily life (2018) - 

Pareto and the famous 20/80 of static unequal distributions of inequality become dynamic Talebian way of thinking and are transformed in an opposite 80/20 answer.

Moreover, during life, the gaps between the predictions of inequality and reality in all models of communist thinking show very large differences between point estimates and temporal distributions, but also in relation to reality. Pragmatic or Talebian in the problem of models of economic well-being and inequalities typologies according to religions, there are associations that demonstrate that Protestantism and Catholicism still brought the greatest well-being...

In my opinion, the article is good, and publishable at any time, but with a few small additions, that is, it only requires a minor revision (some nuanced additions I have tried to detail above). Final answer is 

Accept after minor revision

Reviewer 2 Report

The author presents a study comparing an equivalent market exchange with a redistribution based on power centers and a nonequivalent exchange with mutual aid. The manuscript is suitable for publication after a couple of minor revisions:

(i) As the author defines min and max in line 201, equations (4a) and (4b) sound repetitive. However, metrically, the difference max - min can be replaced simply by |m_i(t) - m_j(t)|. Therefore, I suggest that the author define something like delta = |m_i(t) - m_j(t)|, replacing "max-min" by delta in equations (4c), (4d), (4e), (4f), line 235, (6) for conciseness.

(ii) In Equation (5a), the function Sort() needs to be clarified because the output depends on t and i. If we have a list of m_1(t),...,m_N(t), the Sort(m_i(t)) function should return an ordered list. Thus, I needed to figure out why r_i(t) is a scalar instead an ordered list. Consider the standard order statistics notation as, for example, m_{(1)}(t) \leq ... \leq m_{(N)}(t).
